# Same Viewpoint Different Perspectives—A Comparison of Expert Ratings with a TLS Derived Forest Stand Structural Complexity Index

**Julian Frey [1],\*, Bettina Joa [2], Ulrich Schraml [3] and Barbara Koch [1]**

1   Chair of Remote Sensing and Landscape Information Systems FeLIS, University of Freiburg,
    79106 Freiburg, Germany; Barbara.koch@felis.uni-freiburg.de
2   Faculty of Environment and Natural Resources, University of Freiburg, 79106 Freiburg, Germany;
    Bettina.joa@confobi.uni-freiburg.de
3   Forest Research Institute Baden-Wuerttemberg, 79100 Freiburg, Germany; Ulrich.Schraml@Forst.bwl.de
\*   Correspondence: Julian.frey@felis.uni-freiburg.de; Tel.: +49-761-203-96854

**Abstract:** Forests are one of the most important terrestrial ecosystems for the protection of biodiversity, but at the same time they are under heavy production pressures. In many cases, management optimized for timber production leads to a simplification of forest structures, which is associated with species loss. In recent decades, the concept of retention forestry has been implemented in many parts of the world to mitigate this loss, by increasing structure in managed stands. Although this concept is widely adapted, our understanding what forest structure is and how to reliably measure and quantify it is still lacking. Thus, more insights into the assessment of biodiversity-relevant structures are needed, when aiming to implement retention practices in forest management to reach ambitious conservation goals. In this study we compare expert ratings on forest structural richness with a modern light detection and ranging (LiDAR) -based index, based on 52 research sites, where terrestrial laser scanning (TLS) data and 360° photos have been taken. Using an online survey (n = 444) with interactive 360° panoramic image viewers, we sought to investigate expert opinions on forest structure and learn to what degree measures of structure from terrestrial laser scans mirror experts' estimates. We found that the experts' ratings have large standard deviance and therefore little agreement. Nevertheless, when averaging the large number of participants, they distinguish stands according to their structural richness significantly. The stand structural complexity index (SSCI) was computed for each site from the LiDAR scan data, and this was shown to reflect some of the variation of expert ratings (p = 0.02). Together with covariates describing participants' personal background, image properties and terrain variables, we reached a conditional R² of 0.44 using a linear mixed effect model. The education of the participants had no influence on their ratings, but practical experience showed a clear effect. Because the SSCI and expert opinion align to a significant degree, we conclude that the SSCI is a valuable tool to support forest managers in the selection of retention patches.

**Keywords:** Terrestrial laser scanning; stand structural complexity index; forest structures; retention forestry; guideline implementation; photo-based expert survey; mixed methods;

## 1. Introduction

### 1.1. Relevance of Forest Structure

Forests hold the majority of the global biodiversity [1] and, at the same time, are under heavy use for timber production in several regions, which leads to a strong simplification of their structures

[2]. In conventionally managed forests optimized for timber production, structures typical of late successional stages and old growth forests are especially lacking. This loss of structural diversity leads to strong negative impacts on various ecosystem services like biodiversity and resilience [3]. Changes in forest structure due to wood production are different than changes in structure due to natural disturbance, both in frequency and extent [4]. To minimize this difference, the retention of diverse forest structures has become a common goal in forestry conservation guidelines [5,6]. Retention practices include setting aside single or small groups of "habitat trees" to grow old, the enrichment of standing and lying dead wood within stands, as well as a variety of other measures depending on the ecosystem and local conditions.

### 1.2. Descriptors of Forest Structure

Various authors stress the point that forest structure is the main driver of habitat provision and therefore for forest biodiversity [7–9], but the quantification of forest structures is still a matter of current research, since the direct measurement of biodiversity is costly with current methods. Forest structure has multiple aspects and a broad variety of indicators exist to describe these for retention purposes. While forest managers mostly have to rely on their subjective visual impression when they decide which trees or group of trees will be retained [10], different criteria have been evaluated to make these decisions more objective. First attempts to describe quantitative structural complexity used diversities of tree diameters at breast height (DBH), tree size differences, or spatial patterns of tree locations [11–15]. To our knowledge, none of these have been proven relevant indicators for biodiversity. Newer approaches focus on old growth attributes from manual assessments. These tree-related microhabitats (TreMs), such as cavities or dead branches have proven to be significant habitat descriptors for the diversity of various taxa [5], but suffer from a relatively strong observer bias [16].

Latest indices based on terrestrial laser scanning (TLS) use the detailed geometric description these devices offer to derive indices based on the distribution of material in 3D-space directly. While a lot of research has been conducted to receive classical forest parameters like DBH or crown base height from TLS, indices describing the complexity of the forest directly from the data are very rare. One index based on a single scan which can be acquired in some minutes with minimal expertise and minimal bias from the inventorying person is the Stand Structural Complexity Index (SSCI, [17], see method section for details).

Since those technical measurements of forest structure have yet to be implemented, management guidelines and retention programs try to deliver clear criteria for the selection of retention patches or habitat trees [6,8]. However, several other criteria such as timber yields, aesthetics or accessibility may influence decisions on which trees to mark as habitat trees. The emphasis on economic targets especially limits the amount of discretion that foresters perceive when implementing nature conservation programs [18]. Against the backdrop of a multitude of potentially conflicting goals, choices of retention elements may in many cases be suboptimal for the protection of biodiversity and other ecosystem services [19]. The choice of a retention patch is a complex multidimensional decision, which must consider, among other factors, the rarity of the habitat in its surrounding landscape, the occurrence of rare species, and TreMs into account, factors which are more easily observed by an expert than in a technical index like the SSCI, which is purely focused on the stand. To our knowledge there are no scientific studies, which compare how consistently forest experts rate a forest habitat according to its structural richness and value in terms of biodiversity conservation, and how these ratings mirror or complement technical indices such as the SSCI.

### 1.3. Photography-Based Expert Evaluations

Photo-based studies are commonly used in urban and landscape planning to evaluate the visual perception or beauty of landscapes. In our study, this aspect is of minor importance, since we ask experts to rate forest structural richness with regard to biodiversity conservation, which is not directly linked to aesthetical aspects [20–22]. Even if the complex visual perception biases the expert rating, this is not the focus of our research. Based on an eye-tracking study with 28 participants assessing images of partial retention in recently logged forests, Pihel et al. [23] researched if experts

and lay people rate forest biodiversity differently. Their results indicated that lay people judge forest biodiversity similar to experts, but with less confidence. Furthermore, the eye-tracking information showed significantly different results, indicating that experts have a different way of assessing the forest.

The best impression of a forest stand can be perceived by walking through it. Ideally all senses would be used to experience a forest, but sight is arguably the most important for structural evaluation. Photographs can give us a visual representation of the forest but limit us in our ability to take different viewpoints, perspectives or in focusing on small elements due to resolution constraints. On the other hand, photograph-based surveys have some major advantages against on-site visits. Online photography assessments can reach a large number of participants, which would require large logistical efforts if conducted as on-site visits. Numerous on-site visits may also change or damage the site during the study and site effects like rain, wind, temperature, vegetation change, mosquitos, cloudiness and shades are not easily excludable [24].

While the change of viewpoints by free movement cannot be easily simulated using photographs, 360° panoramic images offer the possibility to choose one's own viewing angle using interactive photo viewers, where the image is projected inside a virtual sphere. The participant can freely look up in the canopy or down to the forest floor, and even zoom on smaller details, which is of major importance in our case, since relevant structures can be found in all parts of the forest. To our knowledge, this is the first 360° image-based online survey conducted in forest research, but the methodology is similar to classical photo surveys [20,21].

In the present study we combined photo-based expert ratings with the new possibilities of TLS-based forest structure quantification. This approach can provide insights on the validity of the structural index, following a similar approach as Ashcroft et al. [25], and help us to discuss the uncertainties in expert ratings in this field. Our main research questions therefore are: (1) Do experts' rate the structural richness of a forest patch consistently? (2) Do technical indices reflect expert opinions? We therefore approached forest experts in a quantitative online survey with 360°-image spheres showing forest research plots where we have taken scans, to derive technical indices and compare the results (see method description for further details). As far as we know such a comparison of technical remote-sensing techniques and expert opinions has not been conducted yet. We hypothesize, that the educational and personal background of the participants, the properties of the forest sites (terrain height and slope, forest type) as well as of the images (brightness, greenness) have an influence on the expert ratings. We further verify if the structural richness is correlated to the habitat quality of a forest according to the expert ratings.

## 2. Methods

The interdisciplinary design of the study integrates various methods and data sources. We applied TLS and a photo based expert survey for data collection. These two datasets were combined with additional information from forest inventory data and a terrain model. Figure 1 gives an overview of the applied processing procedure from prior fieldwork to the final model. The detailed workflow and indices used are described below.

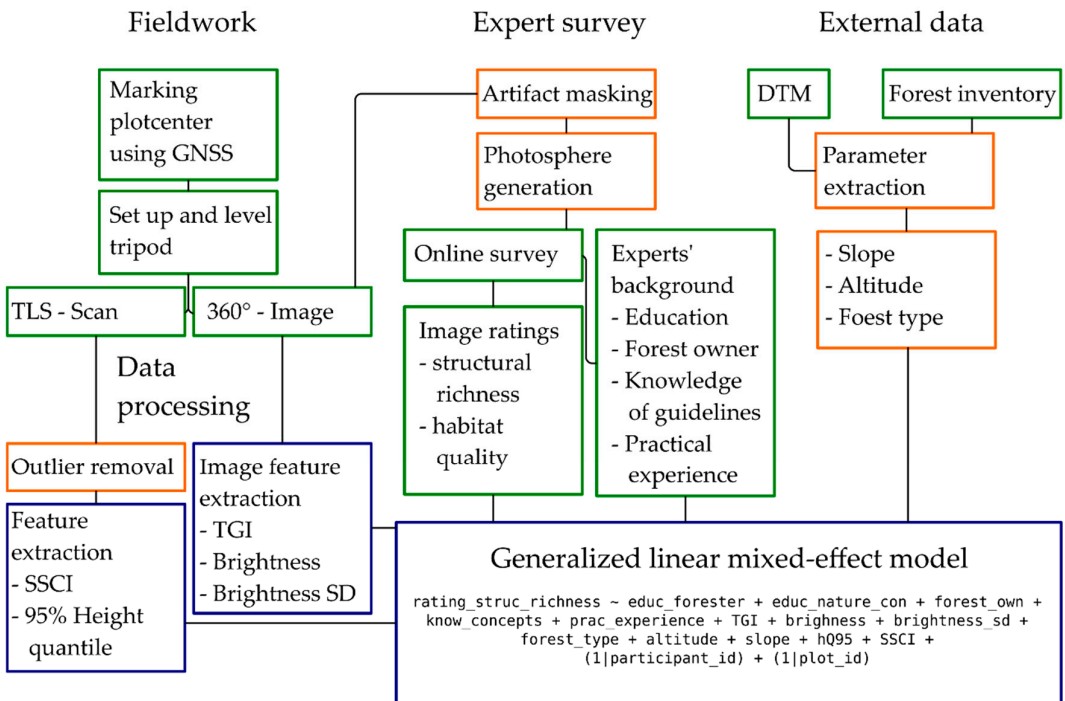

**Figure 1.** Detailed workflow of the whole study. Details about the processing steps can be found in chapters 2.2–2.6. Green boxes indicate data acquisition, orange boxes data preparation and blue boxes data analysis steps. DTM stands for digital terrain model, TGI for triangular greenness index and brightness standard deviation (SD) for the area weighted standard deviance of the image brightness.

### 2.1. Research Site

The study area is located in southwest Germany in the southern Black Forest mountain range in the state of Baden-Württemberg (Figure 2). The Black Forest rises from the Rhine valley up to ca. 1500 m a.s.l. The research project "Conservation of Forest Biodiversity in Multiple-Use Landscapes of Central Europe" (ConFoBi) established a network of one-hectare plots in state-owned forests. Spanning 3700 km², all plots have a minimum distance of 750 m from each other. Forests in this area are dominated by Norway spruce (*Picea abies* L.), European beech (*Fagus sylvatica* L.) and silver fir (*Abies alba* Mill.). To implement the goals of the state level nature conservation strategy [26] in forest management, the so-called old- and deadwood program [27] sets specific targets for minimum amounts of deadwood and habitat trees to be retained in state forests. Management of these forests follows a "close-to-nature" paradigm characterized by single tree and group selection harvests, natural regeneration, promotion of mixed and uneven-aged stands, and retention of habitat trees [28]. The plots were selected so that water bodies, roads, power lines etc. were excluded but smaller man-made objects like raised hides for hunting, skid tracks, hiking paths etc. could be included. Data for our study were collected from 52 plots (Figure 2), which cover a range of altitudes between 502 m and 1295 m a.s.l. and the average plot slope varies between 2° and 34°.

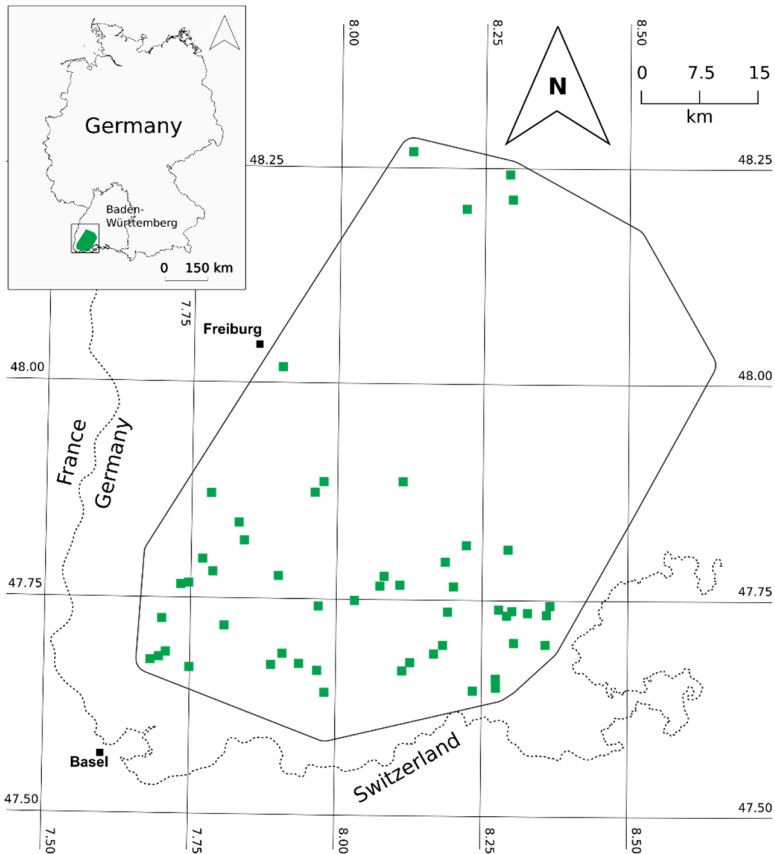

**Figure 2.** Map of the research area within Germany (green in top-left corner), and the location of plots where photos and scans have been taken (green squares).

## 2.2. Terrestrial Laser Scanning and 360° Photographs

All scans and photos were taken in April and May 2018. For every plot, the center point has been marked with a metal pin using a real-time kinematic global navigation satellite system. Above these center points we placed a tripod of approximately 1.3 m height and leveled its top using a bubble level. We first took a terrestrial laser scan of the scene using a Faro Focus 3D 120 TLS (Faro Technologies Inc., Lake Mary, USA, see Figure 3B). The scanner was set to an angular resolution of 0.044°, covering all 360° horizontally and 300° vertically, leaving out a 60° cone at the bottom where the tripod stands. This led to a maximum of 29 million distance measurements translated into a 3D point cloud (Figure 3C). The scanner automatically corrects for small tilts and its rotation using internal sensors. Instrument heights, date and time, GPS-location and qualitative weather information were recorded as metadata for every scan.

We choose a 360° Garmin VIRB 360 action cam (Garmin Europe Ltd., Southampton, Great Britain, see Figure 3A), which consists of two RGB-sensors with fisheye lenses pointing horizontally in opposite directions. The images are automatically stitched together to a full 360 × 360° photosphere by the camera (5640 × 2820 Pixels; 15.90 Megapixels). After we removed the scanner, we placed the camera on the same tripod, to have the best possible consistency in perspective between the images and the scan (see Figure 3D for an overlay of the distance image from the scanner and the 360° photo). After moving out of sight, the field researchers released the camera remotely using a field tablet and collected metadata analogous to the scan.

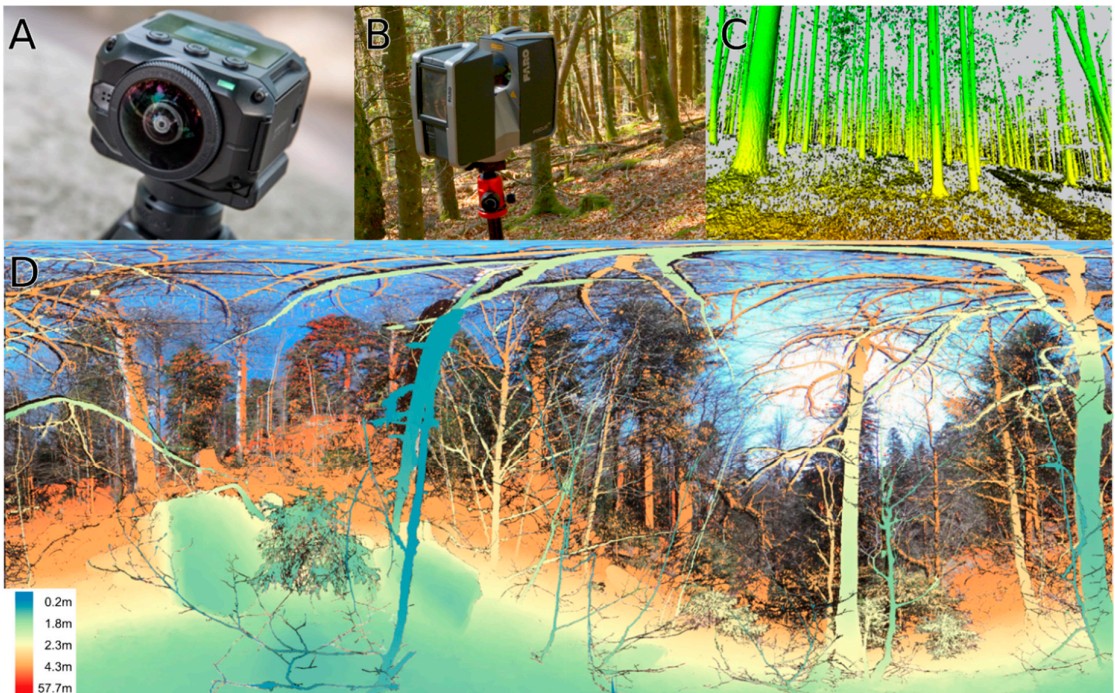

**Figure 3.** (**A**): Garmin VIRB 360 digital camera; (**B**) Faro Focus 120 terrestrial laser scanner; (**C**): Visualization of a point cloud; 2D: overlay of the 360°-image from the camera with the distance image from the scanner - blue colors indicate closer red colors more distant points.

### 2.3. Processing of the Images and Point Clouds

All raw data from the scanner was imported into Faro Scene (v 6.2.4.30) and noise was removed by applying the standard outlier removal with default parameters. All further processing was done in R (version 3.5.0, R Core Team, 2017) after normalizing the point clouds using a one-meter resolution LiDAR-based terrain model provided by a state agency [29].

We computed the SSCI as described by Ehbrecht et al. [17]. The index describes the complexity of the open space between the material in a forest by connecting the points of one vertical planar scanline to a polygon and calculating the area (A) to perimeter (P) ratio called the fractional dimension (FRAC, see Equation (1), Figure 4). Afterward it averages this shape complexity ratio over all scanlines. The resulting mean FRAC is a dimensionless ratio without an absolute scale, which is a drawback in forest where we expect an older and therefore mostly taller forest to be more complex. It is therefore scaled by a layering complexity index (effective number of layers, ENL) based on the ratio of occupied voxels (p) in one-meter thick layers (i) (see Equation (2) [17,30]).

$$SSCI = mean\left(\frac{2 * ln(0.25 * P)}{ln(A)}\right)^{ln(ENL)} \qquad (1)$$

$$ENL = 1\bigg/ \sum_{i=1}^{i\,top} p_i^2 \qquad (2)$$

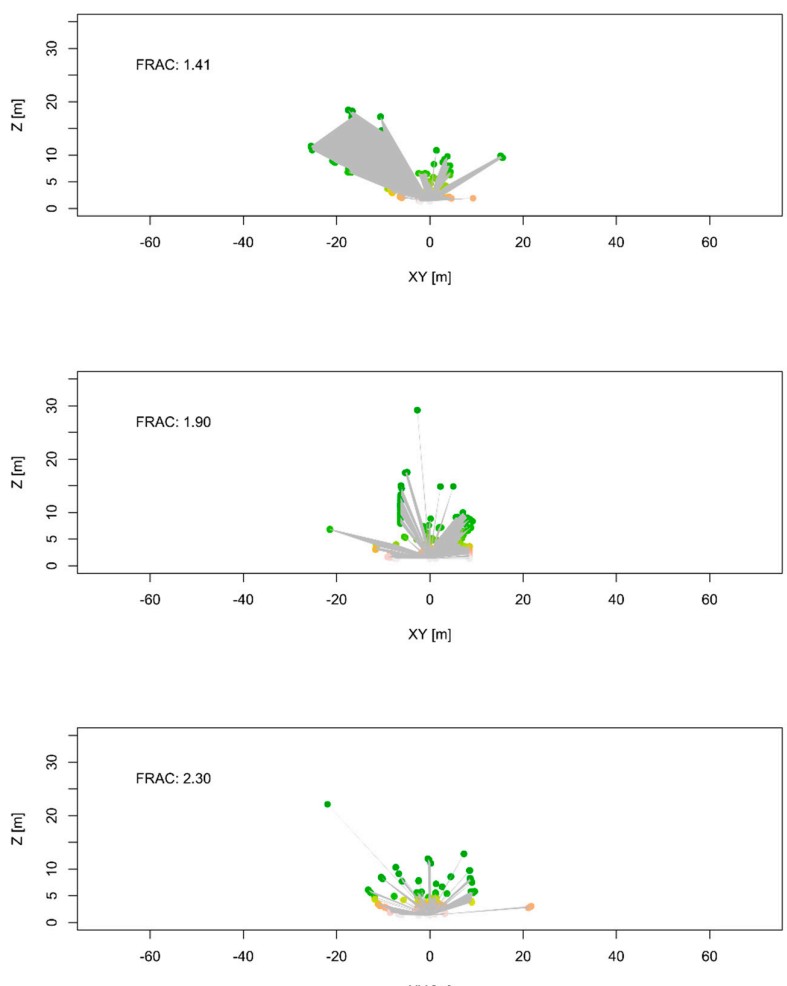

**Figure 4.** Examples of different vertical scanlines from a terrestrial laser scanning (TLS) data for stand structural complexity index (SSCI) calculation. Colored points indicate the returns to the scanner; gray area shows the constructed polygon. The upper figure is an example for low, the central figure for medium and the lower figure for high fractional dimension (FRAC) values.

We chose the SSCI because it is, unlike indices relying on classical forest parameters (such as stem positions or DBH), always computable. The possibility to derive those other indices might be limited by the complexity of the understory and by more complicated shapes of trees. Additionally it is to our knowledge the only index available describing the complexity of distribution of material in the forest besides the ENL, which is already included in the SSCI. All other indices based on LiDAR data require spatially homogenous dense point clouds. As the density of points diminishes with increasing distance to the scanner, those indices are not applicable to our dataset.

To save computation time we thinned the scans by every second row and column. Additionally, we calculated the 95% quantile of the point heights as an indicator for the tree heights. The actual tree height is only roughly determinable because of occlusion effects and probable outliers. The 95% quantile therefore is a relatively robust indicator for the stand height.

For the image spheres, we removed the lower 60° cone where mostly the tripod was visible and replaced it with a white area. Since there were still some plot marks visible, we overlaid these with white circles to make sure no one could determine the actual location of the plot, and to avoid distraction from the task. Next, the images were grouped into three equally sized groups (high, medium, low structure) according to their SSCI rank order derived from the scans (see Figure 5 for

examples). This grouping was done solely to ensure variability in the images an expert was presented in the survey (see next section for details).

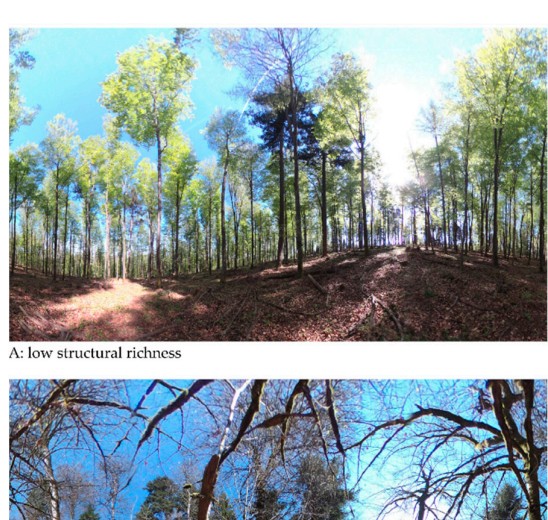

A: low structural richness

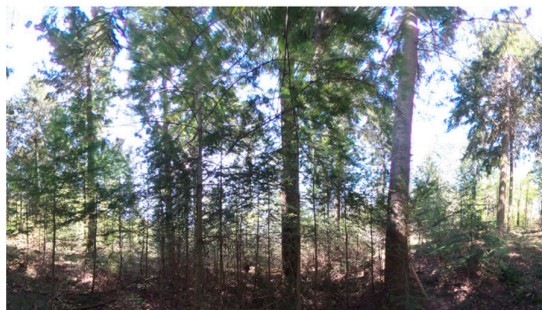

B: medium structural richness

C: high structural richness

**Figure 5.** 360°-images of the calibration phase of the questionnaire, which have been projected to the spherical image viewer. Here depicted in planar Mercator projection for better visibility. The high, medium and low structural richness has been derived according to the SSCI rank order.

### 2.4. Forest Expert Survey

To investigate experts' ratings of structural richness we developed a quantitative survey targeting both researchers and practitioners engaged in the field of forestry or forest conservation (collectively referred to as 'forest experts' within this study). More precisely, this includes foresters, scientists of forestry and ecology, private forest owners, forest auditors, forest and nature conservation administrators and members of environmental organizations. An extensive internet search provided us with ~1250 email addresses, to which we sent survey invitations. We used self-selection sampling and sent an email to all German institutions, associations and administrations that fit our target group, inviting their members to participate in our online survey. Additionally, we distributed advertising postcards at forest related conferences and asked colleagues to distribute our invitation via various channels, to reach a wide variety of participants and gain a maximum number of responses.

The online survey was conducted in German; however, the questions cited in this paper have been translated to English. The survey started with an introduction about our research topic and area,

and an instruction about how to use the survey tool, including the image viewer. In order to reconstruct the experts' rating in their daily working practice best possibly, no rating criteria were specified. The introduction solely pointed to the importance of structural richness for forest biodiversity and referred to the corresponding guidelines.

The questionnaire (see Supplementary material) was divided into two sections: in the first section we showed the participants nine different images and asked them the same two questions for each photo: (1) 'How high would you rate the structural richness of this forest?' and (2) 'How do you rate the ecological value of the recognizable structures with regard to biodiversity conservation?'. Both questions were rated on a five-point Likert scale, ranging from very low to very high structural richness/ecological value, with an additional option 'I cannot rate this'.

The first three images (Figure 5) were identical for all participants, showing the forest site with the second lowest, the second highest and the median SSCI-value to give an impression of the range of the forests' structural richness. We choose the second highest and lowest SSCI, because we wanted to avoid very special outliers. We used these three images in the 'calibration phase', which was excluded from the statistical analysis. The following six images were randomly selected by the survey system (uniform random choice) with the precondition that two images were shown from each of the predefined categories (high, medium, low structure according to SSCI). This design was chosen to ensure that participants are provided a sufficient range of structural richness. Neither the three groups nor the calibration phase were identifiable to the participants, to avoid any bias. The photos were presented in a scroll- and zoomable photosphere viewer to enable participants to choose their own perspectives and zoom in at details.

In the second part of the questionnaire, we asked if the participants know any guidelines/concepts addressing these structures and if they are experienced in selecting habitat trees. The questionnaire ended by asking about professional education (in forestry and/or conservation), private forest ownership and an open comment field.

The questionnaire was pretested at various stages of development. To make questions as understandable and precise as possible, six colleagues with varying levels of forestry and conservation education backgrounds were asked to answer the questionnaire. Afterwards they were interviewed about their understanding and interpretation of the questions. Based on these insights, the questionnaire was revised and implemented online. Another twelve pretesters, including foresters as well as forest and ecological scientists, completed the online version and gave feedback both on the questions' comprehensibility and the functionality and usability of the online survey tool. We adapted the survey accordingly and allowed functionality in the questionnaire to return to previous questions, since various pretesters demanded a possibility to revise previous ratings. The survey was accessible between late September 2018 and early January 2019.

### 2.5. Covariates and Statistical Analysis

We used linear modeling to verify the influence of the SSCI on the expert ratings. According to our hypothesis that an expert's professional background would have an influence on their ratings of structural richness, we included in the model answers to questions regarding the participant's education in forestry or nature conservation, if s/he is a forest owner, familiar with retention forestry concepts, and if the person has practical experience in the assignment of habitat trees. We hypothesized that the plots with relatively high and/or steep terrain, which are not equally common in Germany and therefore not equally familiar to all experts, may lead to varying ratings. Therefore, we included terrain information such as terrain altitude and slope of the plot.

Additionally, we included the 95% quantile of point heights and the main forest type from a full forest inventory of trees above 7 cm DBH in the classes broadleaved, mixed or coniferous. A plot was classified as pure broadleaved or coniferous if 90% of the total basal area of the 1ha-plot correspond to the class, following the definition of the German national forest inventory [31].

It was not possible to acquire the images under controlled lighting conditions with fixed camera settings, therefore we could not standardize image exposure as an influencing factor. We calculated the average brightness, the standard deviance of brightness and the triangular greenness index (TGI)

for the images, which is the area of the triangle defined by the reflectance signals of the visual bands. The index is regarded as an indicator for the chlorophyll content of leaves [32]. As the images for the online survey were projected into a virtual sphere, we calculated the mean values and standard deviations weighted according to the area proportions a pixel occupies on the virtual sphere, and this too was included in the model.

*2.6. Modeling*

We used the lme4 package [33] in R v. 3.5.0 [34]. The three images in the calibration phase were excluded from analysis. All answers which were rated with 'I cannot rate this' were excluded and treated as no reply given. First, we tested if the rating of the images showed significantly different results using a two-sided t-test on all combinations of two images to verify that the ratings differentiate the plots. Additionally, we calculated mean values and standard deviations of the ratings on a plot level. In a second step, we tested if the rating of structural richness directly correlated to the rating of the ecological value of the respective structures.

All covariates were tested for collinearity using the corrplot package [35]. No combination of covariates exceeded a correlation factor of 0.5 and we did not exclude any. The cross over nested design of the experiment with covariates for the single participants as for the plots and images as well, made a linear mixed effect model with two random variables necessary; therefore, we used the lmer function from the lme4 package in R [33]. We formulated the main model to include the participant ID and the plot ID as random effects, the participants' rating of the structural richness as response, and all other mentioned variables as predictors. We repeated the model as a generalized additive model to check for nonlinear effects. The quality of the model improved only slightly, so we continued with the simpler linear version. After visually confirming the normal distribution of the residuals and the random effects using quantile-quantile plots, we computed analysis of variance (ANOVA) on the model. To discuss the explanatory power of the SSCI itself we repeated the model once excluding the SSCI.

## 3. Results

The online survey was begun by 555 of the participants who gave at least one answer. During the calibration phase, 48 of them dropped out. From the remaining 492 participants, 444 completed the survey (reached the last page). We only considered completed questionnaires in our analysis. As not all participants filled all response fields, the n-value is less than 444 in some of the results below. Table 1 describes several characteristics of the participating experts. A clear majority of them have a forestry education background (88%), knowledge of retention forestry guidelines (95%) and practical experience in the selection of habitat trees (84%).

**Table 1.** Characteristics of respondents.

| Attributes of Respondents | | | n |
|---|---|---|---|
| Forest ownership | yes | 131 | 436 |
| | no | 305 | |
| Knowledge of retention forestry guidelines | yes | 413 | 436 |
| | no | 23 | |
| Practical experience in the selection of habitat trees | yes | 369 | 440 |
| | no | 71 | |
| Formal education in forestry | yes | 392 | 436 |
| | no | 44 | |
| Formal education in ecology or nature conservation | yes | 166 | 424 |
| | no | 258 | |
| Age of participants | Min | 19 | 425 |
| | Max | 85 | |
| | Mean | 49 | |

The images showed a sufficient range of structural richness in forests to enable the experts to differentiate the single plots. When testing the 1176 possible combinations of two plots using the t-test, 88% of them are rated significantly different by the participants. The rating of structural richness and ecological value of the respective structures were highly significantly correlated ($p < 0.01$, $R^2 = 0.53$). We proceeded considering only the ratings on structural richness, since this is more closely related to what the SSCI describes. The mean values on plot level varied between 1.7 ('rather low') in a monolayered regrowth stand with coniferous trees and 4.2 ('rather high') in a steep multilayered mixed forest with larger gaps and clearly visible lying and standing dead wood (see Appendix A for the images). The standard deviations of all ratings on one plot varied between 0.6 and 1.0, which is sufficient for some stands to fill nearly the complete five-point Likert scale with its 95% confidence intervals.

The results of the main model can be found in Table 2 and in Figure 6. Experts' practical experience in the selection of habitat trees showed a significant effect in the main model ($p = 0.02$). Contrary to that, the other predictors describing the participants' background showed no significant influence. Neither the forestry education, the knowledge of guidelines nor the education in nature conservation or forest ownership influenced the rating significantly (Table 2). According to the plot terrain and vegetation properties, only the vegetation height indicated by the 95% height quantile from the TLS data ($p = 0.02$), and the SSCI ($p = 0.02$) influenced the rating significantly. The image brightness showed surprisingly very little influence, but the greenness indicated by the TGI was highly significant ($p < 0.01$, Figure 6).

**Table 2.** Model parameter estimates, standard deviations (SD) and p-values.

| Variable | Estimate | SD | p-Value |
|---|---|---|---|
| Intercept | −1.55 | 2.37 | 0.51 |
| Forest ownership [Yes/No] | −0.05/−0.11 | 0.28/0.28 | 0.41 |
| Knowledge of guidelines [Yes/No] | 0.13/0.07 | 0.23/0.25 | 0.74 |
| Practical experience [Yes/No] | −0.26/−0.08 | 0.43/0.44 | **0.02** |
| Educ. in forestry | 0.06/−0.06 | 0.27/0.28 | 0.37 |
| Educ. in ecology or nature conservation [Yes/No] | 0.02/0.07 | 0.13/0.12 | 0.59 |
| SSCI | 0.93 | 0.37 | **0.02** |
| TGI | 0.05 | 0.01 | **<0.01** |
| Brightness | <0.01 | 0.01 | 0.59 |
| Brightness SD | 0.03 | 0.02 | 0.17 |
| Forest type mixed | 0.18 | 0.17 | 0.30 |
| 95% height quantile [m] | 0.04 | 0.02 | **0.03** |
| Slope [deg.] | $4 \times 10^{-3}$ | $8 \times 10^{-3}$ | 0.60 |
| Altitude [km] | −0.33 | 0.34 | 0.33 |
| Observations | | 2637 | |
| N participants | | 444 | |
| N plots | | 49 | |
| Marginal R² | | 0.12 | |
| Conditional R² | | 0.44 | |

For mixed-effects models, $R^2$ can be categorized into two types. Marginal $R^2$ represents the variance explained by fixed factors, while conditional $R^2$ is interpreted as variance explained by both fixed and random factors (i.e., the entire model) [36]. With a conditional $R^2$ of 0.44 the variables show an acceptable explanatory power for the expert rating, but the marginal $R^2$ is low (0.12). That means that the random effects (participant ID and plot ID) largely contribute to the explanatory power of the model. Excluding the SSCI from the model lowers the marginal $R^2$ by 0.02.

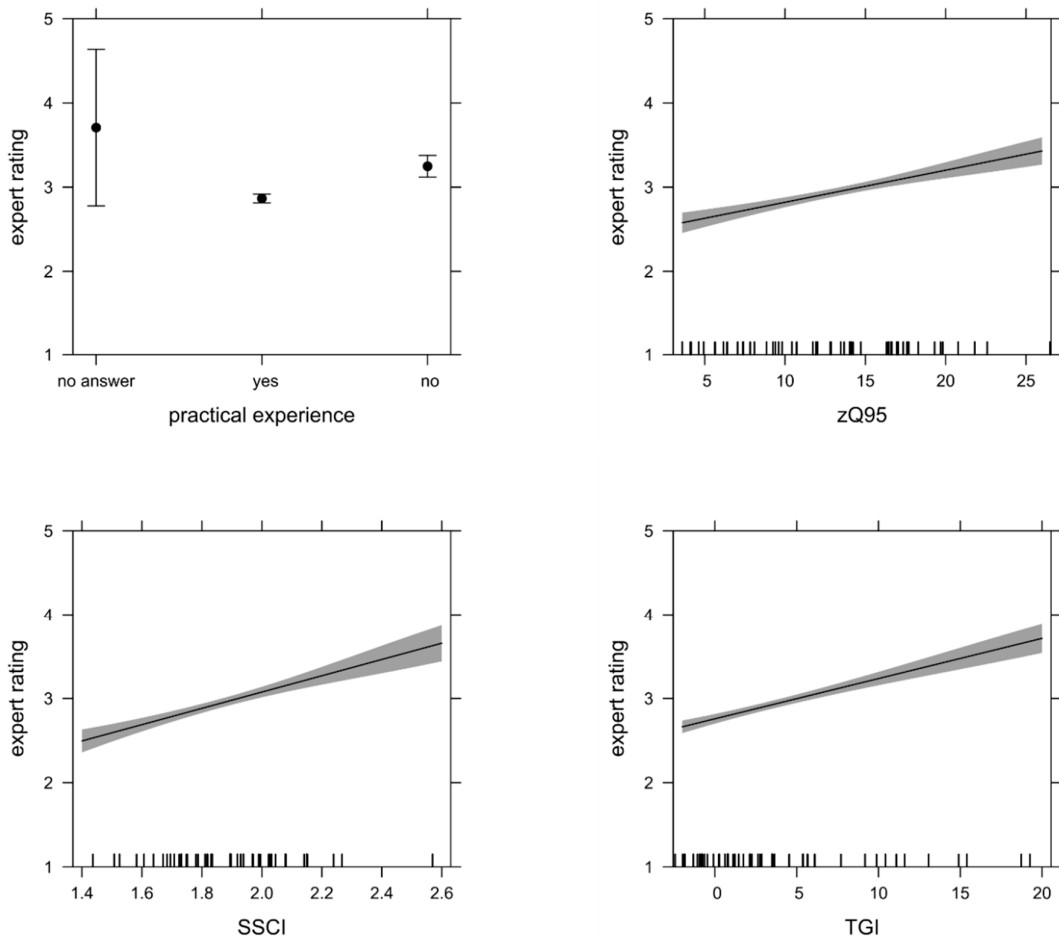

**Figure 6.** Effect plots of the statistically significant predictors. Grey areas and whiskers indicate 95% confidence intervals. The scale from the expert rating goes from 1 very low up to 5 very high structural richness. Upper left: effect of practical experience in the assignment of habitat trees, top right: effect of the 95% height quantile from the TLS data, lower left: effect of the stand structural complexity index (SSCI), lower right: effect of the triangular greenness index (TGI).

## 4. Discussion

### *4.1. Innovations*

The SSCI corresponded to the ratings of experts. This shows on the one hand that the expert ratings were nonrandom and on the other hand, that the SSCI can explain at least some of the structural attributes perceived as relevant by forest experts, who, after all, are the ones who implement biodiversity conservation objectives in forest management. Ashcroft et al. [26] have also chosen an expert approach to verify a TLS-based index on vertical vegetation density profiles, but in a qualitative manner with only three observers. They found a general agreement of the TLS data with the observers, even if the comparison was weak in some respects because the observers estimated the heights poorly. Our results clearly indicate that expert surveys can be a valid method to verify forest structural indices, if other methods are not available or not applicable due to their destructive manner. The large variation in expert ratings suggests that quantitative approaches are preferable for such a task.

The evaluation of forest structures is a multidimensional challenge combining numerous aspects that are not easy to represent technically, which makes automation difficult. On the contrary,

evaluations by humans are always subjective and reflect the subject's specific perspectives. Recent studies indicate that several factors, like education, or familiar management techniques and economic goals are inconsistently influencing tree selection practices, which may also largely depend on individual preferences [19,37,38]. Our finding that greener forests are perceived as richer in structure underlines the hypothesis that even forest experts unconsciously include a variety of factors in their rating of forest structures that may have no major relevance to it. Greenness may indicate biomass productivity, which is however not necessarily linked to structural richness. It may also be linked to the forest type, since broadleaved trees may appear greener, but these predictors were only slightly correlated (r = −0.26). Lastly, the greenness may indicate more light on the forest floor, which would result in a more vital (greener) understory. Future research is required to clarify why experts rate structural richness of greener forests higher and if the TGI relates to other indicators assessing structural richness. The large standard deviations in the expert ratings show that there is no consistent expert agreement on the quantification of forest structure. A study of Cosyns et al. [19] on full inventoried tree-marking training sites has shown that forestry trainers made decisions on habitat tree selection that are more consistent than foresters' or students' choices, which supports the impact of participants' expertise on tree-selection. Nevertheless the overall agreement on which trees to select was low for all groups, which demonstrates the downside that this effect of expertise is limited. The finding that forest trainers' decisions are more consistent is underlined by our result that practical experience in the marking of habitat trees had a significant effect on the rating and that the standard deviations of ratings of participants with practical experience where slightly lower (yes: 0.434 , no: 0.439). The low overall agreement on selected trees found by Cosyns et al. is supported by our results that being formally educated in forestry or nature conservation had no significant effect.

The positive influence of the vegetation height on the structural richness rating in our study is not surprising, since larger trees are related to a higher number of old growth structures [39]. Yet, to us, the fact that the image brightness had no significant influence on the rating was remarkable. We hypothesized that the more bright and 'friendly' images would be rated higher in terms of structural richness, as it was the case for the greener ones.

Another unexpected finding of the study was the terrain variables did not show any significant influence, although we expected the rarer steep habitats to be rated higher in terms of structural richness by the experts.

### 4.2. Reasonability of the Methods and Study Limitations

A photo-based study of walks through different habitats by lay people and experts also revealed no clear aesthetic preference of the participants to more biodiverse landscapes [22]. Our findings are consistent with that study, which may explain why the aesthetic aspect of these images did not influence our experts' rating that was focused on biodiversity-relevant forest structures. The standard deviation of image brightness could have been an indicator for a more heterogeneous light regime on the forest floor, which has been named as a structural attribute by various authors [40–42]. While canopy gaps are regarded as structural elements [41], the SSCI does not cover this aspect well, since gaps do not deliver any TLS points and thus do not increase the complexity aspect of the index. We expected the standard deviation of brightness to cover this factor, as it describes image heterogeneity, which may be related to the forest structural heterogeneity at the same time, but it was not statistically significant.

A weakness of our study is the insufficient resolution of the images that makes it difficult to see smaller details even on more distant trees or in the canopy, as well as the limitation to a single perspective on every stand, which cannot replace a real walk through a forest, which offers multiple perspectives. The photosphere viewer offered more viewing angles than a planar photo, which is an improvement on the limitations of most photo-based questionnaires. In contrast to guided or camera forest walks, the advantage of our approach is, besides reaching a large number of participants from all over Germany, that all of them faced the same conditions according to the weather and vegetation period. Another crucial advantage to this method is that photos can cover a wide range of different forest types, which is rarely accomplishable in a single walk.

Our research area covers only a single mountain range in southwest Germany, which is neither representative nor does it cover the full spectrum of German forests. We tried to address this limitation in the survey introduction text by specifying our specific research area. In the questionnaire, we included calibrating images and gave participants the chance to adapt their rating according to images shown later. Since several experts addressed this shortcoming in the open comment field, we assume that they were aware of this limitation and considered it in their rating. Despite this, the experts distinguished the different stands significantly, which shows that we at least cover some range of forest type found in Germany. We have tried to ensure the variability of images shown to one participant using a grouping based on the SSCI, which is also one of our main predictors. As the participants were not aware about this procedure and thus had no incentive to conform to our pre-selection, we do not expect any bias by this design. The very coarse grouping of just three equally sized groups anyways did not reflect the SSCI values very well, but was able to avoid that participants only got confronted with very similar forest stands by purely random selecting images. This could have led to participants adjusting the range of their rating according to the limited number of images presented. Confronting all participants with all images would have led to a very long survey and respectively high dropout rates or bored random choices.

### 4.3. Data Aquisition and Availability

Terrestrial laser scanners are becoming cheaper, easy to use, and therefore more available for the application in forest management. An increasing amount of software becomes available to analyze TLS datasets, which enables practitioners to use the data of such sensors. We used an additional fine scale digital terrain model (DTM), which is nowadays available for most countries. However, the low explanatory power of the terrain variables showed that such a model is not even necessary especially since the local DTM could also be derived by the laser scans. To our knowledge, there is no common agreement on the setup of the scanning design [30], but at least some specifications (minimum distance to the next tree, height above ground, vegetation phase) may be beneficial to gain comparable results. Further research needs to be done to directly detect and quantify additional structural features like canopy gaps [43], or TreMs [44], that are not ideally represented by the SSCI.

### 5. Conclusion

We found significant relationships between a very technical approach and the complex human perception of forest structures. In combination with standardized guidelines of expert assessment, terrestrial laser scanning can be a valuable objective tool for the quantification of small-scale forest structures. Its abilities as a fast and easy-to-use tool make it a perfect choice for productive and cost-efficient assessments. The high-resolution scans make changes in structural richness quantifiable without relying on the same person to be available for inventories in the following years, which is a striking advantage for monitoring purposes. The disadvantage is that remote sensing is not able to detect small-scale structures reliably on whole trees yet, which would be an important feature to consider within an index. TLS-based indices have not yet shown their ability to quantify habitat quality for a wide range of taxa, which is an important missing step to be addressed in future research. An inventorying person may not only take the richness of structures into account, but also the habitat value those structures provide, which TLS-based indices cannot do. Yet, stands perceived as structurally rich were also rated as highly valuable for biodiversity conservation, and vice versa, which has still not been proven in scientific research. Taking all the considerations into account, and because these two variables were strongly correlated in the experts' rating, we do not consider the lack of habitat quality assessment as a major drawback for TLS in our study.

If one takes a closer look at the difficulty of assessing forest habitat sites with their numerous interacting and influencing factors, it becomes clear that a purely technical assessment can hardly account for all relevant factors. Laser scanning must, therefore, be regarded as a supporting tool which cannot replace human evaluation, but it may complement, support and improve it. Since it is still in question if indices are indeed sensitive to structures supporting biodiversity in general or just

for certain taxa, we should attempt to use the best and most general index available, as long as full biodiversity inventories are highly cost intensive.

**Supplementary Materials:** The full questionnaire translated into English is available online at www.mdpi.com/xxx/s1. A live preview of the image sphere viewer including images from the survey is online available under: http://omnibus.uni-freiburg.de/~jf1050

**Author contributions:** Conceptualization, J.F. and B.J.; Methodology, B.J. and J.F.; Software, J.F.; Validation, B.J. and J.F.; Formal Analysis, J.F. and B.J.; Resources, B.K. and U.S.; Data Curation, J.F. and B.J.; Writing – Original Draft Preparation, J.F.; Writing – Review and Editing, J.F., B.J., U.S. and B.K..; Visualization, J.F.; Supervision, B.K. and U.S.; Project Administration, J.F. and B.J.; Funding Acquisition, B.K. and U.S.

**Founding**: This study was funded by the German Research Foundation (DFG), ConFoBi grant number GRK 2123/1 TPX. The article processing charge was funded by the DFG and the University of Freiburg in the funding program Open Access Publishing.

**Acknowledgments:**The authors warmly thank all survey participants, the whole ConFoBi-Team for the help and the fruitful discussions, especially Carsten Dormann for all of his expertise, Martin Denter for his contribution during the fieldwork and Linda Weber for data preparation. We also thank Taylor Shaw for her extensive proofreading. We thank the State Agency for Spatial Information and Rural Development of Baden-Württemberg (LGL) for the provision of the digital terrain model.

**Conflicts of Interest:** The authors declare no conflict of interest.

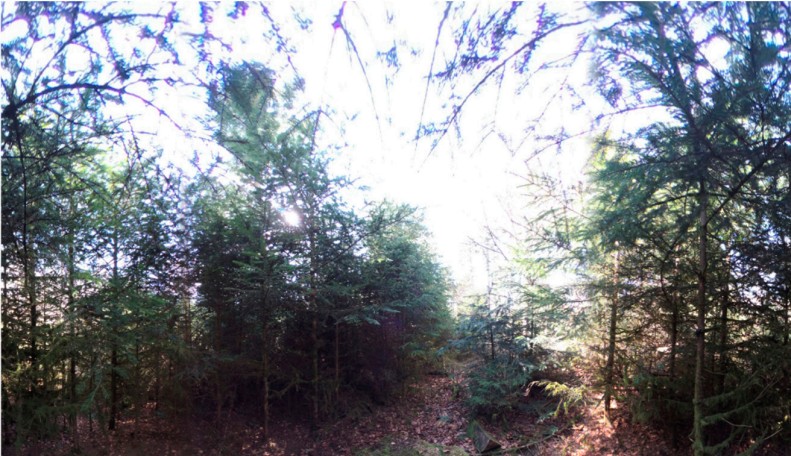

A: lowest structural richness

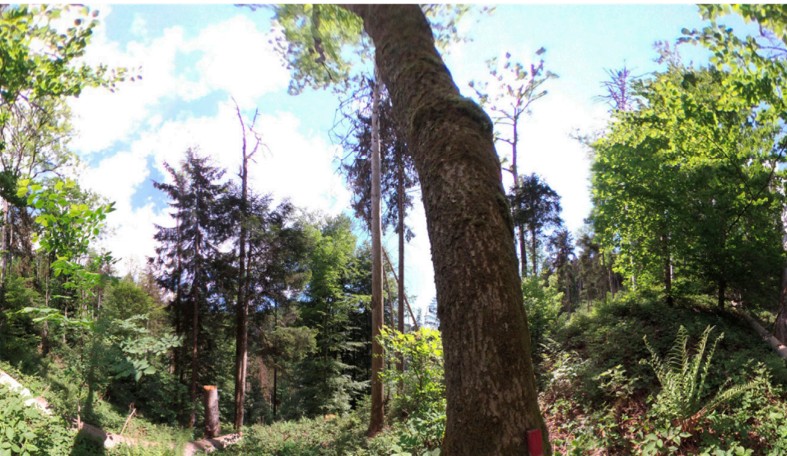

B: highest structural richness

**Figure A1.** Images with the lowest (**A**) and the highest (**B**) structural richness according to expert ratings.

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
