# Peer review of "Same Viewpoint Different Perspectives—A Comparison of Expert Ratings with a TLS Derived Forest Stand Structural Complexity Index"

_remotesensing, doi:10.3390/rs11091137_

Round 1

Reviewer 1 Report

This manuscript provides quite a different method to thoroughly describe the human perception of forest structures by using the LiDAR-based cloud and 360-degree photos. I think it merits publication after a major revision. The shortcomings were listed as follows,

1. At the beginning of section 2,  I strongly suggest the authors add an overall technical chart with a description of the methods and processing procedures for the field survey and data acquired.

2. In section 2.5, the authors should explain why linear modeling was employed to verify the influence of the SSCI on the expert ratings? Alternatively, does a nonlinear model sound better?

3. In lines 289-290, please note there are prerequisites for performing the ANOVA test. You need carefully check the data distribution and confirm whether the normal distribution based ANOVA is more fitable than the non-parametric ANOVA, known as the Kruskal-Wallis H test.

4. The interpretations of the results were somewhat too simple and need
more detailed information.
5. The discussion section needs to be reorganized into several subsections, focusing on data availability, reasonability of the methods, innovations/new findings, and limitation.

6. Please remove the citation numbers of the references. I think the conclusion should merely be the summarized section of your original idea from the study.

Author Response

Dear Reviewer,

Thank you very much for your very constructive and helpful contributions to our work. We hope that our changes based on your suggestions are able to convince you. Your comments helped us to better structure our work and to make several points more clear.

Please find all detailed changes below.

With kind regards,

Julian Frey and Bettina Joa

1. At the beginning of section 2,  I strongly suggest the authors add an overall technical chart with a description of the methods and processing procedures for the field survey and data acquired.

                We have implemented the figure as proposed and an introductory paragraph for the chapter. (l.129)

2. In section 2.5, the authors should explain why linear modeling was employed to verify the influence of the SSCI on the expert ratings? Alternatively, does a nonlinear model sound better?

                I am usually in favor of GAM and we also tested them using the GAMM4 package. The results are basically the same (significance levels change slightly for the anyway significant variables). We have therefore opted for the simpler and more widespread method, since the method part in this interdisciplinary work is already very extensive and GAMs would need some additional explanation. We have now mentioned this in the later section 2.6 about the model itself: “We have also repeated the model as Generalized Additive Model to check for nonlinear effects. The quality of the model has improved only slightly, so we have continued with the simpler linear version.” (l. 301)

3. In lines 289-290, please note there are prerequisites for performing the ANOVA test. You need carefully check the data distribution and confirm whether the normal distribution based ANOVA is more fitable than the non-parametric ANOVA, known as the Kruskal-Wallis H test.

                I am sorry that this was not clear enough; we have performed the ANOVA on the model and not on the data itself. We confirmed normal distribution of the residuals of the whole model, as well as of the random effects visually using q-q-plots. We have added this information in the respective paragraph.  (l. 303)

4. The interpretations of the results were somewhat too simple and need more detailed information.

                We have added Table 2 with more detailed information about the model output in the results section (l. 337) and made several additions and improvements in the discussion section (l. 350).

5. The discussion section needs to be reorganized into several subsections, focusing on data availability, reasonability of the methods, innovations/new findings, and limitation.

                We have restructured the discussion section in line with the proposal (l. 350).

6. Please remove the citation numbers of the references. I think the conclusion should merely be the summarized section of your original idea from the study.

                We assume that the comment refers to the conclusion. We have removed the citations or moved them to the discussion, and added a sentence in the beginning answering our main research question (l. 454).

Reviewer 2 Report

The manuscript 493647 by Frey et al. focuses on the comparison between expert ratings on forest structural richness derived from photographs and a terrestrial laser scanning based index of stand structural complexity called SSCI.

The authors put together a very interesting and solid manuscript. It is well structured, concise and the findings are certainly worth being published. I have only a very few issues that should be addressed.

Minor points:

Abstract L.16-18 and elsewhere: Please consider that you refer to landscape structure a lot (e.g. also at l.82; l.90). Your study addressed plot level structure and I think there is a scaling problem if we want to get from plot level structural complexity to landscape complexity. Schall et al. 2018 “The impact of even‐aged and uneven‐aged forest management on regional biodiversity of multiple taxa in European beech forests.” made this clear. Your methods look at plot scale structural complexity, which is fine, but please consider revising or making more careful statements towards landscape level.

l.109, l.382: I suggest to make a better distinction between “perspective”, which is actually a specific point from which a scene is looked at, and the “viewing angle”. The latter enables for different views from the same perspective. The words are used in a confusing and partly misleading way. Consider revision. Should be a gib deal.

l.197: Did you adopt this 95% quantile approach from ALS-research or can you provide a source indicating that this is actually a solid estimate of stand height from TLS data?

Figure 5: It would be nice to see the data points in the plots.

Results: Did I miss it or is there actually no overview on the explanatory power of the individual variables? Would be nice to know how they perfomered individually. Is that possible?

l.373: In the SSCI gaps are “overjumped” in the cross-sections as nicely shown in Figure 3 top (upper left area of the polygon). Doesn’t that mean that gaps are what they are: no structural “object” but actually the absence of structures.  I would disagree that this means that they do not contribute to the complexity aspect of the index. I would agree though that they do not increase complexity. Consider revision of the sentence to make this clearer.

Figure 5: Figure captions should be understandable without reading the text. Therefore I suggest to provide the SSCI and TGI in full, not only the abbreviations. Especially, since TGI is used only once more in the manuscript.

l.349-351: I disagree that “greenness” of a forest may have no major relevance to the structures. In fact, I would assume that a greener forest is likely greener as it is has higher stand structural complexity. Please provide arguments for your hypothesis. Generally it would be nice to know more about this TGI as it seemed to be a good predictor!?

Author Response

Dear Reviewer,

Thank you very much for your very constructive and helpful contributions to our work. We hope that our changes based on your suggestions are able to convince you. We are convinced that your contributions have significantly improved our work. 

Please find all detailed changes below.

With kind regards,

Julian Frey and Bettina Joa

Abstract L.16-18 and elsewhere: Please consider that you refer to landscape structure a lot (e.g. also at l.82; l.90). Your study addressed plot level structure and I think there is a scaling problem if we want to get from plot level structural complexity to landscape complexity. Schall et al. 2018 “The impact of even‐aged and uneven‐aged forest management on regional biodiversity of multiple taxa in European beech forests.” made this clear. Your methods look at plot scale structural complexity, which is fine, but please consider revising or making more careful statements towards landscape level.

                We have adapted our wording accordingly. (l.18, l.83,l.91)

l.109, l.382: I suggest to make a better distinction between “perspective”, which is actually a specific point from which a scene is looked at, and the “viewing angle”. The latter enables for different views from the same perspective. The words are used in a confusing and partly misleading way. Consider revision. Should be a gib deal.

                Changed as proposed.

l.197: Did you adopt this 95% quantile approach from ALS-research or can you provide a source indicating that this is actually a solid estimate of stand height from TLS data?

                We have adapted this approach from ALS research, especially from biomass estimations. To my knowledge there is no literature using a quantile approach for the determination of relative stand heights. We decided to use this approach because the strong occlusion effects in single scans make it quite unlikely to hit the highest spot of a tree crown. Giving absolute heights could therefore be easily misinterpreted. Various authors have tackled the retrieving of forest inventory variables from TLS (e.g. Moskal and Zheng 2012, https://doi.org/10.3390/rs4010001)   

Figure 5: It would be nice to see the data points in the plots.

                Due to the large variance in the expert ratings this looks quite messy and limits the readability of the plots. Therefore, we would prefer to keep them as they are. (l.344)

Results: Did I miss it or is there actually no overview on the explanatory power of the individual variables? Would be nice to know how they perfomered individually. Is that possible?

                We have added Table 2 as you proposed. (l. 315)

l.373: In the SSCI gaps are “overjumped” in the cross-sections as nicely shown in Figure 3 top (upper left area of the polygon). Doesn’t that mean that gaps are what they are: no structural “object” but actually the absence of structures.  I would disagree that this means that they do not contribute to the complexity aspect of the index. I would agree though that they do not increase complexity. Consider revision of the sentence to make this clearer.

                Changed as proposed (l. 413).

Figure 5: Figure captions should be understandable without reading the text. Therefore I suggest to provide the SSCI and TGI in full, not only the abbreviations. Especially, since TGI is used only once more in the manuscript.

                We have added the written out versions of the abbreviations in the caption to the figure. (l.345)

l.349-351: I disagree that “greenness” of a forest may have no major relevance to the structures. In fact, I would assume that a greener forest is likely greener as it is has higher stand structural complexity. Please provide arguments for your hypothesis. Generally it would be nice to know more about this TGI as it seemed to be a good predictor!?

                We are not able using our data to distinguish between an indication of forest structure and a bias from visual preferences. We have discussed this in more detail know in the respective part of the discussion (l. 368). Additionally, we have added a very brief explanation of the index in the respective methods section (l. 282) and removed the sentence about it from the abstract (l.34).

Round 2

Reviewer 1 Report

This latest version of the manuscript shows the better quality, and I recommend it for publication in its present form.